# Analysis of Torque and Force Induced by Rotary Nickel-Titanium Instruments during Root Canal Preparation: A Systematic Review

**Myint Thu [1], Arata Ebihara [1,*], Sherif Adel [1,2] and Takashi Okiji [1]**

1 Department of Pulp Biology and Endodontics, Division of Oral Health Sciences, Graduate School of Medical and Dental Sciences, Tokyo Medical and Dental University, Tokyo 113-8549, Japan; thuendo@tmd.ac.jp (M.T.); adelendo@tmd.ac.jp (S.A.); t.okiji.endo@tmd.ac.jp (T.O.)
2 Department of Restorative and Dental Materials, Oral and Dental Research Division, National Research Centre of Egypt, Cairo 12622, Egypt
* Correspondence: a.ebihara.endo@tmd.ac.jp

**Abstract:** The aim of this review was to provide a detailed literature analysis of torque and force generation during nickel-titanium rotary root canal instrumentation. We followed Preferred Reporting Items for Systematic Reviews and Meta-Analyses (PRISMA) guidelines. An electronic search was performed using in PubMed and in journals for articles published in English from 1987 to June 2020 on studies that investigated dynamic torque and force in vivo or in vitro. We assessed article titles and abstracts to remove duplicates, and the titles and abstracts of the remaining articles were screened for eligibility. Full texts were read to verify eligibility by considering predetermined inclusion and exclusion criteria. Fifty-two out of 4096 studies met the inclusion criteria, from which we identified 26 factors that influence torque or force generation. Factors associated with higher torque or force generation and supported by multiple studies with mostly consistent results included convex triangle cross-sectional design, regressive taper, short pitch length, large instrument size, small canal size, single-length preparation technique, long preparation time, deep insertion depth, low rate of insertion, continuous rotation (torque), reciprocating motion (force), lower rotational speed and conventional alloy. However, several factors are interrelated, which obscured the independent effect of each factor, and there was insufficient scientific evidence supporting the influence of some factors.

**Keywords:** dynamic torsional test; force; nickel-titanium rotary instrument; root canal preparation; screw-in tendency; torque

## 1. Introduction

Nickel–titanium alloy (Ni-Ti) rotary instruments must exert torque to cut and eradicate septic dentin during canal preparation; torsional stress, associated with friction between the instrument and dentin wall, accumulates in the instruments [1,2]. The accumulated stress can be retained as residual stress—plastic deformation—after withdrawal of the instrument from the canal if the stress exceeds the elastic limit [3]. The engagement of rotary instruments, especially those with spiral-shaped active cutting edges, with the dentin wall can generate apically-directed screw-in forces, causing the instrument to become locked in the canal [4]. When this occurs, additional torque is required for the instrument to continue rotating. Thus, torsional stress is instantly accumulated in the instrument, leading to torsional fracture [5,6], which is dissimilar to cyclic fatigue fracture caused by the repeated tension/compression stresses at the curvature [7]. Furthermore, screw-in forces may cause the instrument to engage beyond the apical foramen [8] and result in the extrusion of microbes into periapical tissue [9], root weakening, and cracks in the apical area [10].

Numerous studies have been conducted to examine the dynamic torque and force characteristics of Ni-Ti instrument systems to identify factors having an impact on the

stress generated within the rotary instruments. Though potential influencing factors, such as instrument pitch length [11–14] and instrument rake angle [11,15], have been discussed and debated, no single-most important factor has been identified. Thus, there continues to be debate on how the stress generated within Ni-Ti instruments during root canal instrumentation can be limited to a level at which clinical safety is ensured. The aim of this review was to conduct a detailed literature analysis to identify factors governing torque and force generation within Ni-Ti rotary root canal instruments during use.

## 2. Materials and Methods

We followed Preferred Reporting Items for Systematic Reviews and Meta-Analyses PRISMA guidelines [16].

Inclusion criteria:

1.  The article reported torque and/or force in an in vivo study.
2.  The article investigated torque and/or force in an in vitro study using either extracted teeth, cadaveric teeth, plastic/resin blocks, or dentin discs.

Exclusion criteria:

1.  The article examined torque and/or force in a three-dimension finite element study.

An electronic search was carried out in the PubMed database using the following search string: torque and nickel-titanium instruments OR force and nickel-titanium instruments OR torque and load and nickel-titanium instruments OR torque and apical force and nickel-titanium instruments OR torque and vertical force and nickel-titanium instruments OR torque and screw-in force and nickel-titanium instruments OR torque and screw-in effect and nickel-titanium instruments OR torque and screw-in tendency and nickel-titanium instruments OR dynamic torsional test and nickel-titanium instruments. The search was limited to articles published in English. The search date range was set from 1987 to June 2020, to include a period 1 year prior to the first introduction of endodontic Ni-Ti instruments. The same search parameters were applied to search the following journals: Australian Endodontic Journal; Dental Materials Journal; International Endodontic Journal; Journal of Endodontics; Odontology; Oral Surgery, Oral Medicine, Oral Pathology & Oral Radiology, and Restorative Dentistry & Endodontics (formerly Journal of Korean Academy of Conservative Dentistry).

Titles and abstracts were initially evaluated, duplicates were removed, and full texts were read to determine whether articles were eligible based on the inclusion and exclusion criteria. Moreover, for extensive discussion and identification of possible limitations, the studies using various root canal models were included in this review according to the inclusion and exclusion criteria.

## 3. Results

As shown in Figure 1, 4096 articles were identified. After duplicates were removed and preliminary screening was conducted, 75 articles underwent full-text review. Fifty-two studies (Table 1) were eligible for inclusion.

From these studies, we identified the following 26 factors that influence Ni-Ti rotary instrument torque and force: type of sample [17], canal curvature [17,18], cross-sectional design [11,15,19–21], taper [21], blade [15], pitch length [11–15], helix angle [13,15,20], rake angle [11,12,15], cutting efficiency [11,12], instrument size [9,11,12,22–24], glide-path preparation [25,26], canal size [27–31], contact area [32,33], preparation technique [8,23,32–38], preparation time [21,39], insertion depth [17,23,28–38,40–45], insertion rate [41,46], displacement [40], motor [39], kinematics [3,4,20,38,47–52], operative motion [53], rotational speed [36,41], pecking speed [42], lubricant [54,55], experience of the operators [23], and metallurgy [14,20,21,43,44,50,56–59].

Pro-branded systems, such as ProFile, ProTaper Next, ProTaper and ProTaper Universal, were most frequently investigated (Figure 2). The highest numbers of articles were

published in 2019 and in the first half of 2020 (Figure 3), and 48% of the articles included in this review were published in Journal of Endodontics (Figure 4).

The main findings obtained from the present systematic review can be summarized as: Higher torque or force generation was related to convex triangle cross-sectional design, regressive taper, short pitch length, large instrument size, small canal size, single-length preparation technique, long preparation time, deep insertion depth, low rate of insertion, continuous rotation (torque), reciprocating motion (force), lower rotational speed and conventional alloy.

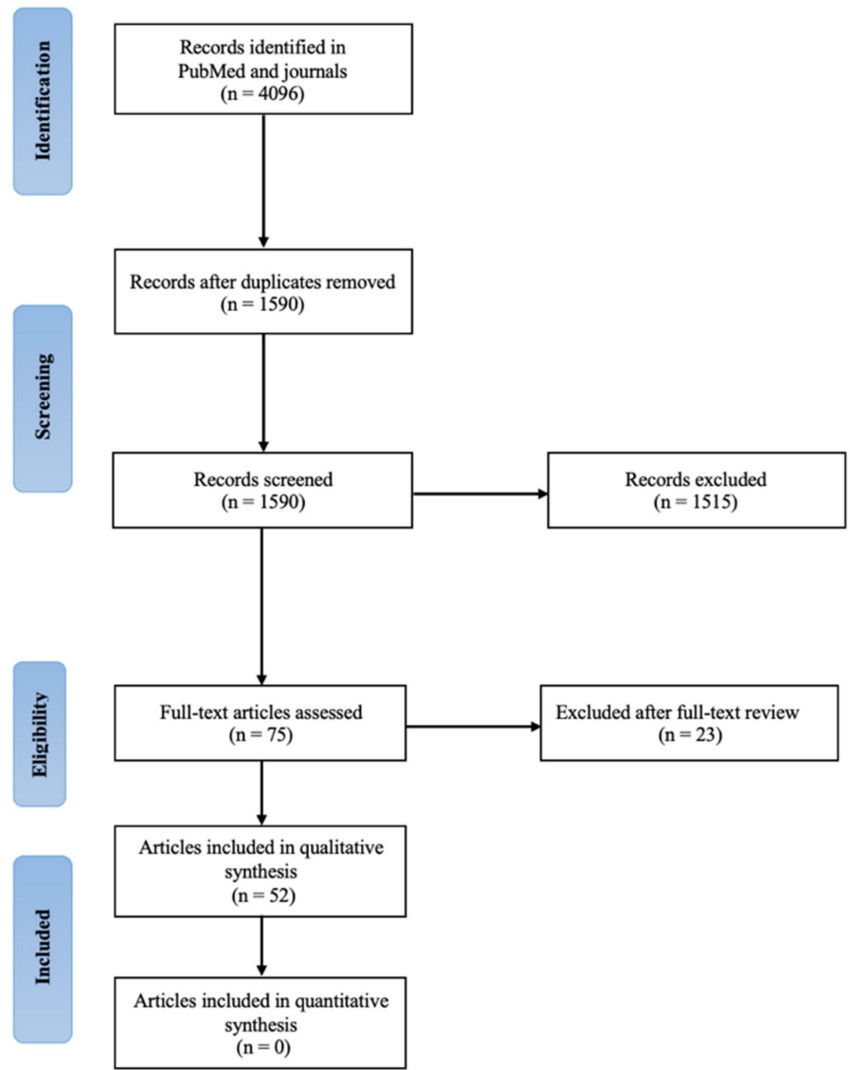

**Figure 1.** Summary of the search processs.

**Table 1.** Summary of reviewed studies analyzing torque and force generated by nickel-titanium rotary instruments during root canal preparation.

| Author/Year | Type of Study | Type of Sample | Instrument | Preparation Technique | Result |
|---|---|---|---|---|---|
| Blum et al., 1999 [23] | In vitro | Mandibular incisors | ProFile | Step-back & crown-down | Students > Endodontist (T)<br>Students < Endodontist (F)<br>Step-back > crown-down |
| Blum et al., 1999 [32] | In vitro | Mandibular incisors | ProFile | Step-back & crown-down | Contact area- 10 mm (Step-back) and 5 mm (crown-down) from the tip<br>Step-back > crown-down (T & F) |
| Sattapan et al., 2000 [28] | In vitro | Maxillary/mandibular central and lateral incisors | Quantec Series 2000 | Single file | Small canal > medium canal |
| Peters et al., 2002 [17] | In vitro | Extracted human teeth/plastic blocks | Profile | Crown-down | Straight canal blocks > curved canal blocks > natural teeth (T)<br>Curved canal blocks > natural teeth > straight canal blocks (F) |
| Peters et al., 2003 [29] | In vitro | Maxillary molars | ProTaper | Single-length | F3 > F2 > F1 > S1 > S2 (T)<br>F3 > S1 > F2 > F1 > S2 (F) |
| Blum et al., 2003 [30] | In vitro | Maxillary central incisors/ mandibular central or lateral incisors | ProTaper | Single-Length | Narrow canal > large canal (T)<br>Large canal > narrow canal (F) |
| Hübscher et al., 2003 [31] | In vitro | Maxillary molars | FlexMaster | Crown-down-like modified sequence | Constricted canal > wide canal (T & F) |
| Diemer et al., 2004 [13] | In vitro | Resin blocks | Hero | Single file | Shorter pitch length > longer pitch length (T & F) |
| Da Silva et al., 2005 [34] | In vitro | Maxillary/mandibular central and lateral incisors | Race 720, Race 721, Profile | Single-length | Profile > Race 720 > Race 721 (T & F) |
| Schrader et al., 2005 [33] | In vitro | Plastic blocks | Profile | Crown-down | 35/0.04 had peak T and F in 4% taper sequence<br>40/0.06 and 35/0.04 had peak T and F in combination of 4% and 6% taper, respectively |
| Peters et al., 2005 [54] | In vitro | Dentin discs | ProFile, ProTaper | Single file | Glyde > Control > EDTA > $H_2O$ for ProFile (T)<br>Glyde > $H_2O$ > EDTA > Control for ProTaper (T)<br>Glyde > $H_2O$ > Control > EDTA for ProFile (F)<br>Control > $H_2O$ > Glyde > EDTA for ProTaper (F)<br>ProTaper > ProFile (T)<br>ProFile > ProTaper (F) |

**Table 1.** *Cont.*

| Author/Year | Type of Study | Type of Sample | Instrument | Preparation Technique | Result |
|---|---|---|---|---|---|
| Boessler et al., 2007 [55] | In vitro | Dentin discs | ProFile | Single file | Dry Control > NaOCL 1% > $H_2O$ > HEPB 18% (T) Dry Control > $H_2O$ > NaOCL 1% > HEPB 18% (F) |
| Boessler et al., 2009 [56] | In vitro | Dentin discs | ProTaper | Single-length | Electropolished > machined (T) Machined > electropolished (F) |
| Diop et al., 2009 [40] | In vitro | Human cadaveric mandivular central/lateral incisors | ProTaper | Single file | Apical > coronal (T & F) Right > left (F) Posterior > anterior (F) |
| Ha et al., 2010 [15] | In vitro | Resin blocks | K3, Mtwo, NRT, ProFile, ProTaper | Single file | ProTaper > K3 > NRT-safe tip > NRT-active tip > Mtwo > ProFile (F) |
| Son et al., 2010 [18] | In vitro | Resin blocks | ProTaper, ProFile | Single-length | $0° > 10° > 20° > 30°$ canal curvature (F) |
| Sung et al., 2010 [24] | In vitro | Resin blocks | ProFile, GT rotary, K3 | Single-length | Greater taper > smaller taper (F) |
| Bardsley et al., 2011 [36] | In vitro | Plastic blocks | Vortex | Crown-down | 200 rpm > 400 rpm > 600 rpm (T & F) |
| Peters et al., 2012 [37] | In vitro | Plastic blocks | Hyflex CM | Single-length & crown-down | Single-length > crown-down (T & F) |
| Ha et al., 2012 [8] | In vitro | Endo-training blocks | PathFile, NiTiFlex, ProTaper | Single file | #13 > #15 > #18 > #20 (T & F) |
| Diemer et al., 2013 [19] | In vitro | Resin blocks | HeroShaper, Prototypes | Single file | H6 > H0 > H4 (T) H0 > H6 > H4 (F) |
| Pereira et al., 2013 [41] | In vitro | Plastic blocks | ProTaper Next | Single-length | 250 rpm > 300 rpm > 350 rpm (T & F) |
| Arias et al., 2014 [27] | In vitro | Maxillary incisors/mandibular molars | ProTaper Next, ProTaper Universal | Single-length | Small canals > large canals (T & F) |
| Pereira et al., 2015 [43] | In vitro | Plastic blocks | ProTaper Universal, Profile Vortex, Vortex Blue, Typhoon Infinite Flex | Single-length | Typhoon > ProTaper Universal > Vortex Blue > ProFile Vortex (T) ProTaper Universal > ProFile Vortex > Vortex Blue > Typhoon (F) |
| Ha et al., 2015 [9] | In vitro | Resin block | G-1, G-2, uG glide path files | Single file | G-2 > uG > G-1 (F) |
| Peixoto et al., 2015 [11] | In vitro | Acrylic blocks | Mtwo, Race, ProTaper Universal | Single file | ProTaper Universal > Race > Mtwo (T) ProTaper Universal > Mtwo > Race (F) |
| Arias et al., 2016 [35] | In vitro | Mandibular molars | PathFile, ProGlider | Single-length & Single file | PathFile 1 > PathFile 2 > ProGlider (T & F) ProGlider (16/.02, single file) > PathFile (16/.02, Sequence) (T & F) |
| Moreinos et al., 2016 [59] | In vitro | Simulated metal block canal | Gentlefile, ProTaper Next, Revo-S | Single-length | ProTaper X1 > Revo-S SC2 > Gentlefile 1 (F) Revo-S SC3 > ProTaper X2 > Gentlefile 2 (F) |

**Table 1.** *Cont.*

| Author/Year | Type of Study | Type of Sample | Instrument | Preparation Technique | Result |
|---|---|---|---|---|---|
| Kwak et al., 2016 [14] | In vitro | Resin blocks | OneG, pG, OneG heat-treated, pG heat-treated glide path files | Single file | pG > OneG > OneG heat-treated > pG heat-treated (F) |
| Ha et al., 2016 [4] | In vitro | Resin blocks | Mtwo, Reciproc 25, ProTaper Universal, ProTaper Next | Single file | Reciproc 25 > ProTaper Universal > ProTaper Next > Mtwo (F) |
| Jamleh et al., 2016 [51] | In vitro | Premolar teeth | Twisted File, Twisted File Adaptive, ProTaper Universal, ProTaper Next | Single-length | ProTaper Universal > ProTaper Next > Twisted File > Twisted File Adaptive (L) |
| Arias et al., 2017 [26] | In vitro | Mandibular molars | PathFile, ProGlider, ProTaper Gold | Single-length | Glide path reduced the torque of shaping files |
| Tokita et al., 2017 [47] | In vitro | Resin canal models | Twisted File Adaptive | Single-length | CR > torque-sensitive reciprocation > time-dependent reciprocation (T) Time-dependent reciprocation > CR > torque-sensitive reciprocation (F) |
| Ha et al., 2017 [45] | In vitro | Resin Canals | One G | Single-length | 4/6 pecking depath > 2/4 pecking depath (F) |
| Fukumori et al., 2018 [22] | In vitro | Resin canals | EndoWave | Single file | EndoWave (30/0.06) > EndoWave (30/0.04) (T & F) |
| Kwak et al., 2018 [25] | In vitro | Resin blocks | WaveOne, WaveOne Gold | Single file | WaveOne > WaveOne Gold (T) Without Glide Path > with Glide Path (T) |
| Jamleh et al., 2018 [58] | In vitro | Maxillary premolar teeth | WaveOne, WaveOne Gold | Single file | WaveOne > WaveOne Gold (F) |
| Nishijo et al., 2018 [50] | In vitro | Endo training blocks | Hyflex EDM Glide Path File (EDM), Hyflex GPF, Scout Race (Race) | Single file | Hyflex EDM Glide Path File > GPF > Race (CR) (F) Hyflex EDM Glide Path File > Race > Hyflex GPF (Reciprocation) (F) |
| Gambarini et al., 2019 [53] | In vitro | Maxillary anterior teeth | Twisted File | Single file | Inward pecking motion > outward brushing motion (T) |
| Abu-Tahun et al., 2019 [46] | In vitro | Resin canals | One G, Hyflex EDM | Single file | No glide path > 5 insertions > 10 insertions > 15 insertions > 20 insertions (T) |
| Kwak et al., 2019 [20] | In vitro | Resin blocks | ProTaper Universal, ProTaper Gold, WaveOne, WaveOne Gold | Single-length & Single file | ProTaper Universal > WaveOne > ProTaper Gold > WaveOne Gold (F) |
| Nayak et al., 2019 [52] | In vitro | Resin blocks | WaveOne Gold, Self-adjusting file, 2Shape | Single-length &single file | WaveOne Gold > 2Shape 2 > 2Shape 1 > self-adjusting file (F) |
| Kwak et al., 2019 [49] | In vitro | Resin blocks | K3XF, Twisted File Adaptive | Single-length | K3XF (CR) > K3XF (adaptive motion) > TFA (adaptive motion) (T) |
| Maki et al., 2019 [42] | In vitro | Resin canal blocks | ProTaper Next | Single-length | High and/or medium-speed > low-speed (clockwise T) High-speed > medium-speed > low-speed (F) |

**Table 1.** *Cont.*

| Author/Year | Type of Study | Type of Sample | Instrument | Preparation Technique | Result |
|---|---|---|---|---|---|
| Gambarini et al., 2019 [21] | In vivo | Double-rooted maxillary premolars | ProTaper Next, EdgeFile | Single-length | ProTaper Next > EdgeFile (T & preparation time) |
| Bürklein et al., 2019 [39] | In vitro | Maxillary incisors | K-flexofile stainless steel, F6 SkyTaper & EndoPilot, DentaPort ZX OTR, VDW.silver | Balanced-force, single-length | Balanced-force > rotary (F) Rotary > balanced-force (T) No significant differences among 3 motors (T) |
| Almeida et al., 2020 [12] | In vitro | Acrylic blocks | ProTaper Next, ProTaper Universal | Single-length | ProTaper Next X2 > ProTaper Next X1 > ProTaper Universal S1 > ProTaper Universal F1 (T) ProTaper Next X1 > ProTaper Universal S2 > ProTaper Next X2 > ProTaper Universal F1 (F) |
| Maki et al., 2020 [57] | In vitro | Resin blocks | Reciproc, Reciproc Blue | Single file | Reciproc > Reciproc Blue (T & F) |
| Lee et al., 2020 [3] | In vitro | Molars | ProTaper Next, One Curve, Hyflex EDM, Twisted File Adaptive | Single-length | CR > adaptive motion (T) Hyflex EDM > One Curve > ProTaper Next > Twisted File Adaptive (T) |
| Htun et al., 2020 [48] | In vitro | Mandibular incisors | Hyflex EDM glide path file, stainless steel K-file | Single file | CR > OGP > stainless steel manual (T in positive domain) CR > stainless steel manual > OGP (T in negative domain) OGP > stainless steel manual > CR (F in positive domain) OGP > CR > stainless steel manual (F in negative domain) |
| Kimura et al., 2020 [38] | In vitro | Resin blocks | Endowave | Single file, crown-down | Single file (CR) > single file (OTR) (clockwise & counterclockwise T) crown-down (CR) > crown-down (OTR) (clockwise T) crown-down (OTR) > crown-down (CR) (counterclockwise T) |
| Peters et al., 2020 [44] | In vitro | Plastic blocks | TruNatomy, ProTaper Next | Single-length | ProTaper Next X2 > ProTaper Next X3 > ProTaper Next X1 > TruNatomy 36 > TruNatomy 26 > TruNatomy 20 (T) ProTaper Next X1 > ProTaper Next X2 > ProTaper Next X3 > TruNatomy 36 > TruNatomy 26 > TruNatomy 20 (F) |

CR: continuous rotation, F: force, L: load, OGP: optimum glide path, OTR: optimum torque reverse, T: torque.

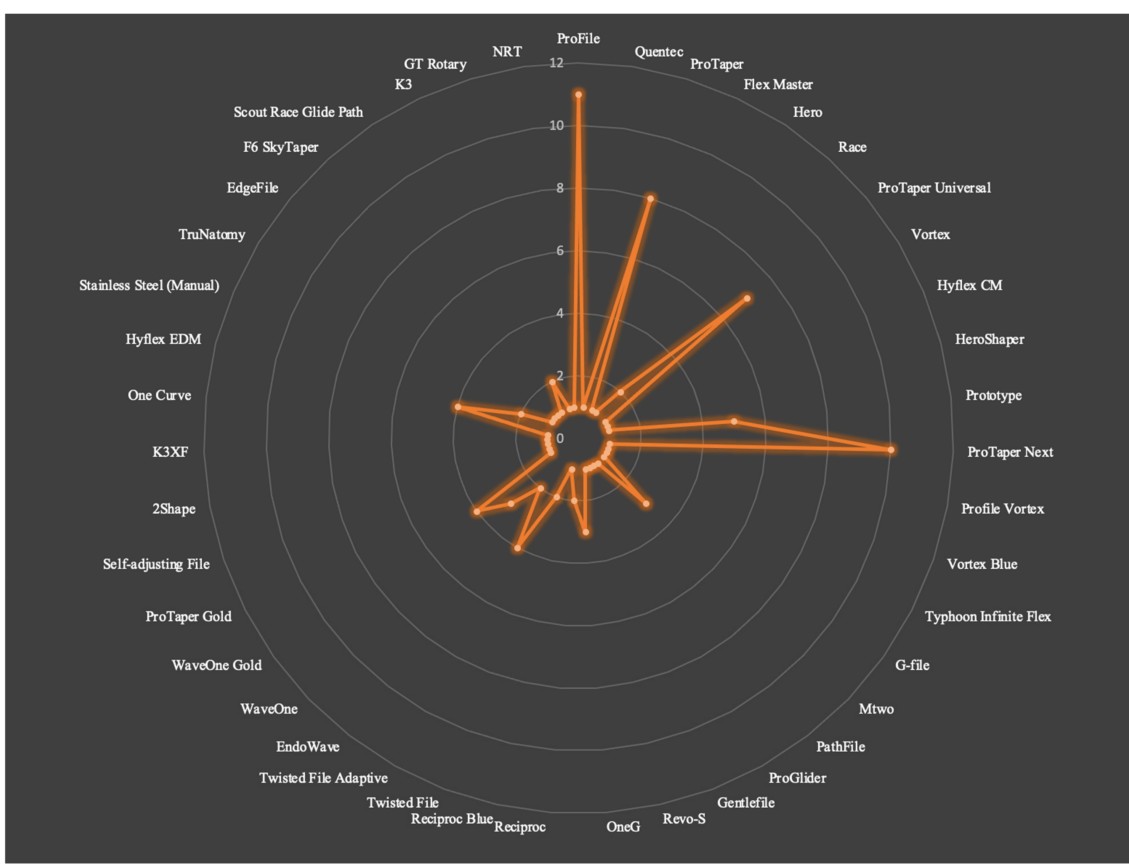

**Figure 2.** Instrument systems used in the studies included in the review.

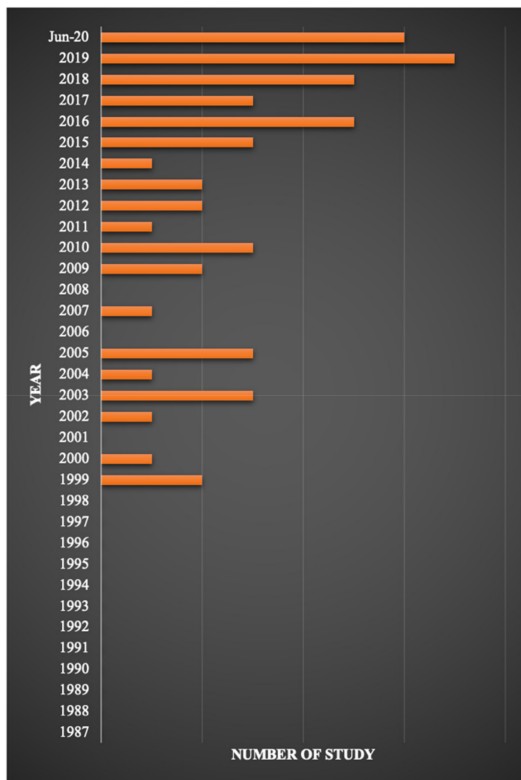

**Figure 3.** Publication year of the articles included in the review.

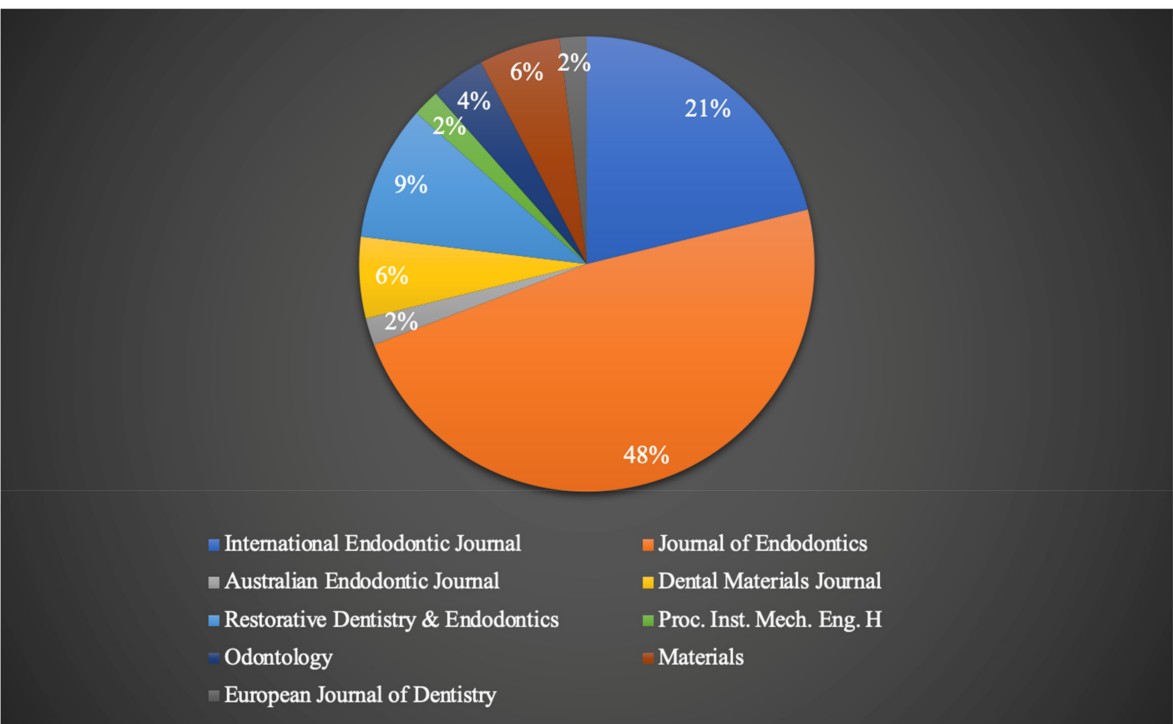

**Figure 4.** Journals in which articles included in the review were published (by percentage). "Proc. Inst. Mech. Eng. H" refers to "Proceedings of the Institution of Mechanical Engineers, Part H: Journal of Engineering in Medicine".

## 4. Discussion

During synthesis, we concentrated on torque and force generated during root canal preparation and their governing factors. Most of the identified factors were interrelated, and thus, the possible inherent factors were extracted from each study and discussed appropriately in the relevant sections.

### 4.1. Type of Sample and Canal Curvature

Although the hardness of dentin is almost twice that of resin blocks [60], instruments exhibited lower torque when used in the canals of single-rooted teeth than when used in resin blocks [17]. The article did not report the sizes of canals in the extracted teeth or whether resin block canal sizes were matched with those of the extracted teeth [17]. In resin blocks, canals are standardized, and the instruments may encounter similar trajectories with similar contact areas, whereas natural canals can be variable in size and configuration, and thus each canal may yield a different trajectory and contact area. Regarding this, the differences in sizes and configurations of the canals may be the matter rather than the differences in surface texture.

In resin blocks, lower torque [17] and lower forces were measured in canals with higher curvature compared to those with low curvature [18]. The influence of the inclination of torque and force vectors in curved canals should be taken into consideration in regard to this matter. This is because the torque and force generated in inclined planes in the curved canals can be lost during measuring, as the transducers only record vertical vectors that represent the values induced in the horizontal plane in the straight part of the root canal [61].

### 4.2. Instrument Design and Cutting Efficiency

In an in vivo study, there was a significantly lower mean torque when a triangular cross-section was used than when an off-centered rectangular cross-section was used [21]. Use of a convex triangle cross-section exhibited the highest torque or force compared with the use of triangle and S-shaped [11]; parallelogram and modified–convex triangle [20];

and triangle, S-shaped, and modified-rectangle cross-sections [15]. Asymmetric triangle cross-sections produced lower forces and similar torque compared with symmetric triangle cross-sections [19].

A constant-tapered instrument exhibited lower torque than a regressive-tapered instrument [21]. Instruments with sharp blades exhibited significantly higher screw-in forces than those with radial lands [15].

Instruments with shorter pitch lengths produced higher torque and higher forces [11–13] because dentin/resin chips can accumulate between the flutes, diminishing cutting efficiency, which may require that greater apical forces be applied [12]. Medium or large pitch lengths produced low torque, which were correlated with better cutting efficiencies [11,12]. However, another study found the opposite—that pitch length did not influence the generation of screw-in forces [14]. Differences in pitch lengths within apical 2 mm of the tip of the instruments were also found to be unlikely to affect screw-in forces [15]. Small helical angles reduced the amount of screw-in force applied and torsional stress generated when instruments with the same size and cross-section were used [13]; however, use of a variable helical angle design did not have an effect on screw-in force [15,20].

In an examination using similar-sized instruments with different cross-sectional designs, higher rake angles (negative) produced higher torque and forces [11]. In another study, however, instruments with positive rake angles produced greater screw-in forces than those with negative rake angles [15]. Instrument size influenced both torque and force when instruments with similar rake angle were tested [12].

Thus, design had an impact on torque and force—shaping instruments with shorter pitch lengths produced greater values, helix angle and rake angle affected the results, and higher cutting efficiencies were associated with lower outcomes. However, care should be taken in interpreting the results of studies that compared instruments with different geometric features because of the inherent limitation that a particular geometric parameter may influence multiple mechanical properties of Ni-Ti rotary instruments [2,11,12,62–68], such as bending, cutting efficiency, and stress and strain distribution pattern, and torque and force generation should be interpreted as the cumulative effect of those influencing factors.

### 4.3. Instrument Size and Canal Size

Based on the articles reviewed, large instruments produced higher torque [12,22,23] and higher apical forces [9,11,22–24] than small counterparts; however, ProTaper NEXT X1 (Dentsply Sirona, Ballaigues, Switzerland) and ProTaper Universal F1 (Dentsply Sirona, Ballaigues, Switzerland) exhibited higher apical forces than the larger-sized counterparts, that is, ProTaper NEXT X2 and ProTaper Universal S2, respectively [12].

Prior glide-path preparation reduced the torque and force exhibited by shaping instruments [25], but using different glide-path instruments did not affect the torque and force exhibited by the shaping instruments [26]. Although several articles stated that the mean torque [27–31] and mean forces [27–29,31] for constricted canals were higher than those for wide canals, one study demonstrated that large canals exhibited higher force in comparison with small canals [30].

### 4.4. Contact Area

According to one study using ProFile instruments (Maillefer, Ballaigues, Switzerland), the contact areas were 1–11 mm from the tip in step-back and crown-down techniques, except for the first three instruments in the crown-down technique, in which 2–3 mm at the apex were devoid of contact [32]; therefore, use of the step-back technique was associated with larger contact areas, generating higher forces and torque compared with the crown-down technique [32]. Conversely, another study found that a larger contact area was correlated with high torque but not with high forces [33].

Available evidence indicates that the contact area varied depending on the length of the instrument inserted in the canals, and thus, the contact area of instruments with various

sizes should be examined at uniform lengths using canals with varying diameter and size because instrument size, canal size, and contact area may be confounding factors.

### 4.5. Preparation Technique

Sequential usage of instruments not only reduced contact areas and improved cutting efficiency [32]; it reduced the torque and force of subsequently used instruments [8,34,35]. When a crown-down technique with a sandwich usage of ProFile instruments (Dentsply Maillefer, Ballaigues, Switzerland) with two different tapers (4% and 6%) was compared with a single-taper technique (4%), the first instrument to the full working length (#35/0.04 taper) exhibited higher torque and force in the single-taper technique and #25/0.06 taper showed the highest force in the combined-taper technique [33].

We described, earlier, that step-back and crown-down techniques generated similar mean torque, whereas a significantly lower mean force was found for the step-back technique than that found for the crown-down technique [23,32]. The mean torque and mean force were likely increased in the apical enlargement and initial preparation stage, respectively, in the crown-down technique [36]. In comparison with single-length technique, almost all instruments used in the crown-down technique exhibited lower torque and force [37]. A similar result was observed during final apical preparation in crown-down technique [38]. Therefore, preparation techniques yielded different results because contact areas and insertion depths vary.

To draw a conclusion from evidence, the crown-down technique exhibits lower torque and force compared with those exhibited by single-length techniques. Force is lower when using step-back techniques than when using crown-down techniques. Based on the studies we reviewed, we suggest that the same preparation technique using different instrument systems should be carried out; there was a lack of comparative examinations of single-length and step-back techniques.

### 4.6. Preparation Time

EdgeFile X7 (Edge Endo; Albuquerque, NM, U.S.) had shorter preparation times and was associated with lower torque than ProTaper NEXT X1 (Dentsply Sirona) [21]. This result was supported by a subsequent study, wherein the torsional resistance and operative torque were tested in parallel, and EdgeFile X7 also exhibited lower torque (wider "torque range" regarded as safer range for clinical use) in comparison with ProTaper NEXT instruments [69]. Manual preparation with a balanced-force technique was significantly slower and exhibited lower torque and higher apical forces than single-length rotary preparation technique [39]. Geometric designs, metallurgy, and preparation techniques should be taken into account when considering preparation time since they determine the cutting efficiency and ability to penetrate the canal, shortening preparation time. Pecking speed and applied force should also be evaluated to find out the correlations with preparation time.

### 4.7. Insertion Depth and Number of Insertions

Several studies have discussed that the deeper the insertion depth of the instrument into the canal, the higher the torque and forces [17,23,28–38,40–45] because of the increased difficulty of the instrument in progressing into the narrow part of the canal [21]. Thus, short pecking depths should be applied in clinical settings with great caution, particularly in the deepest part of the canal. Lower insertion rates produced higher torque and force [41]. Repeated insertion of glide-path instruments tended to lower the torque exhibited by subsequently used shaping instruments [46]. This suggests that repeated insertion reduced the contact area between the canal wall and the subsequent instruments.

### 4.8. Correlation with Displacement

One study [40] analyzed torque about the apical-coronal axis and force in apical–coronal, left–right, and anterior–posterior directions using ProTaper Universal instruments

(Dentsply Sirona) and demonstrated that force values were higher in apical, right, and posterior directions [40]. The hand position and handedness of the operator may have had an effect.

### 4.9. Motor

One study [39] that compared three motors (EndoPilot (Schlumbohm, Brockstedt, Germany); DentaPort ZX OTR (Morita, Osaka, Japan); VDW.silver (VDW, Munich Germany)) using auto torque reverse or optimum torque reverse (OTR) found that, while nonsignificant differences were observed, the DentaPort ZX OTR motor sometimes increased torque in OTR mode [39]. A similar result was obtained for torque-sensitive reciprocal motion [47].

### 4.10. Kinematic

Continuous rotation developed a higher torque than OTR motion during the apical preparation step in the single-length technique [38], torque-sensitive (equivalent to OTR) and time-dependent reciprocating motions [47], optimum glide-path motion and stainless-steel manual preparation [48], and adaptive motion [3,49].

The apical force of continuous rotation was lower than that of OTR used in the crown-down technique, and conversely, apical forces of continuous rotation were higher than those of OTR used in the single-length technique [38]. In addition, continuous rotary glide-path instruments produced higher apical forces than those produced by reciprocating motion [50], stainless steel manual and optimum glide-path motion [48], and adaptive motion [51]; however, Scout Race instrument (FKG Dentaire, La Chaux-de-Fonds, Switzerland) exhibited lower apical force [50]. Time-dependent reciprocating motion exhibited the highest mean apically directed force compared with torque-sensitive reciprocating motion and continuous rotation [47]. However, studies have suggested that, rather than differences in the motion, geometric differences [4] and heated-treated alloy with smaller cross-sectional areas with fewer contact points [20] affect the screw-in force. In those studies [4,20], the screw-in forces produced by reciprocating instruments were higher and lower, respectively, than those by continuous rotating instruments. Transline vibrating motion also exhibited lower forces than those exhibited by continuous rotation and reciprocating motion [52].

Thus, continuous rotation exhibited higher torque than time- or torque-dependent reciprocating motions, and forces differed from motion to motion. Reciprocating motion exhibited higher forces than continuous rotation in most studies because reciprocating instruments can be pushed into the canal during the counterclockwise rotation [38]. Another possible reason could be the aggregate impact of the packing motion and the change in angular momentum between clockwise and counterclockwise rotation (and vice versa), which pushes the instruments down and produces more apically directed force.

### 4.11. Operative Motion

In a study in which in vivo torque generation was compared between inward pecking motions with slight apical pressure and upward lateral brushing motions, the upward brushing motion developed less torque compared with the inward pecking motion [53]. However, it is believed that the upward movements in pecking allow the release of stress in the instrument. Applying upward brushing motions may diminish the stress-releasing mechanism, resulting in more stress accumulation compared with upward pecking motions.

### 4.12. Rotational Speed and Pecking Speed

Torque and apical forces were higher for low rotational speeds than those for higher rotational speeds [36,41]. In the downward direction, ProTaper NEXT instruments (Dentsply Sirona) operated with a high pecking speed (100 mm/min) exhibited higher torque than those operated with medium (50 mm/min) and low (10 mm/min) speeds [42]. A higher apical force was observed with ProTaper NEXT X2 in the high-speed group [42].

### 4.13. Lubricant

Studies showed that lubricant reduced torque and force and that the aqueous type was superior to paste- or gel-type [54,55,70].

### 4.14. Experience

Operator experience was identified as an important factor; torque induced by novice students was significantly higher than that induced by experienced clinicians; the opposite was true for vertical force [23].

### 4.15. Metallurgy

Conventional Ni-Ti instruments produced higher torque or force compared with M-wire, gold wire, blue heat-treated and shape memory alloy instruments [20,43]. Similarly, non-heat-treated glide-path instruments produced higher screw-in forces compared with those produced by heat-treated instruments [14]. However, conventionally machined instruments generated lower torque but higher forces compared with those of their electropolished counterparts [56].

Several studies compared torque and forces of various heat-treated instruments. M-wire instruments (ProTaper NEXT; Dentsply Sirona) generated a higher torque and force than those of postproduction heat-treated Ni-Ti instruments (TruNatomy; Dentsply Sirona) [44]. Other M-wire instruments ((Reciproc; VDW), (WaveOne; Dentsply Sirona) and ProTaper NEXT) had higher torque or force values than blue heat-treated instruments (Reciproc Blue; VDW) [57], gold-wire instruments (WaveOne Gold; Dentsply Sirona) [58], and FireWire instruments (Edge Endo; Albuquerque, NM, U.S.) [21]. One rotary instrument made of stainless steel (Gentlefile, MedicNRG, Kibbutz Afikim, Israel) exhibited a lower force than that exhibited by Ni-Ti rotary instruments ((ProTaper NEXT) and (Revo-S; Micro-Mega, Besancon, France)) [59].

Conventional Ni-Ti alloy instruments exhibited higher torque or apical forces than instruments made of heat-treated alloys in most cases. Instruments made of different alloys with different heat treatments exhibited different torque and forces, and instrument size, design, and kinematics should be standardized to test the abstruse metallurgical properties of different alloys.

The limitation of this review is that there was no uniform standardization of experimental designs, including root canal models and testing devices, which caused methodological heterogeneity among the included studies.

## 5. Conclusions

Among 26 factors identified and analyzed in this study, factors that were associated with higher torque or forces and supported by multiple studies with mostly consistent results included convex triangle cross-sectional instrument design, regressive taper, short pitch length, large instrument size, small canal size, single-length preparation technique, long preparation time, deep insertion, low insertion rate, continuous rotation (torque), reciprocating motion (force), low rotational speed, and conventional Ni-Ti alloy; however, several factors were interrelated, which obscured the independent effect of each. In addition, there was insufficient scientific evidence to support the influence of some factors.

**Author Contributions:** Conceptualization, M.T.; methodology, M.T.; literature research, M.T., S.A.; writing—original draft, M.T.; writing—review and editing, S.A., A.E., T.O.; supervision, A.E., T.O. All authors have read and agreed to the published version of the manuscript.

**Funding:** This research did not receive any specific grant from funding agencies in the public, commercial, or not-for-profit sectors.

**Institutional Review Board Statement:** Not applicable.

**Informed Consent Statement:** Not applicable.

**Data Availability Statement:** Not applicable.

**Conflicts of Interest:** The authors declare no conflict of interest.

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
