# Peer review of "Analysis of Torque and Force Induced by Rotary Nickel-Titanium Instruments during Root Canal Preparation: A Systematic Review"

_applsci, doi:10.3390/app11073079_

Round 1
Reviewer 1 Report
This manuscript is a well analyzed study regarding torque and force generation during Ni-Ti rotary instruments.
Please add some important findings from this study at the end of results.(Higher torque or force generation was related with convex triangle cross-sectional design, regressive taper, short pitch length, large instrument size,,,,etc.)
Please add the limitation of this study. (i.e. You used in vivo, in vitro, and resin blocks studies. There is no uniform standard between various studies)
Reviewer 2 Report
This research is under the scope of this journal; the topic is relevant for readers and this research deals with potentially significant knowledge to the field and opens a new way for future studies. The aim of this paper is quite interesting.
However, there are some concerns about the present manuscript:
Introduction
- The fracture of the instruments in the canal is a current problem in dental activity and increase the difficult case for dental professionals. For this reason, I think is important to distingue the torsion stress from cycle fatigue. Please add a reference. Palma PJ et al. Odontology. 2019 Jul;107(3):324-332. https://doi.org/10.1007/s10266-018-0401-2.
Material and Methods
- When mentioning products/materials or devices: for some of them, you don't mention the manufacturer at all, for some you mention only the manufacturer, for some the manufacturer and city, for some you mention the manufacturer and city/ country.
Reviewer 3 Report
The paper is a systematic review on the Analysis of Torque and Force Induced by Rotary Nickel-Titanium Instruments during Root Canal Preparation.
The authors made a great work in terms of methodology and the paper sounds scientific and very well written.
In the introduction:
- Line 39-40, page 2: “additional torque is required for the instrument to continue rotating, precisely the rest of its part, and torsional stress is instantly accumulated in the instrument” Please explain in a more discursive way “the rest of its part”, and kindly try to make this sentence easier to understand.
- Line 49, page 2: “…has been identified Thus, there continues…” Please correct the punctuation.
In materials and methods, that are clear and well explained:
- Line 60 page 2: “2. The article investigated torque and/or force in an in vitro study using either 3. extracted teeth, cadaveric teeth, plastic/resin blocks, or dentin discs.” Please fix this list.
Results are easy to understand and comprehensive. All the studied characteristics were reported in tables which are clear and concise.
Figure 2: “Instruments systems” Please choose between this or "instrument systems" as written in the text instead.
In Discussion: This section is comprehensive in its various aspects. I suggest improving with an easier to read list:
- 1 Type of Sample and Canal Curvature
- 2 Instrument Design and Cutting Efficiency
- 3 Instrument Size and Canal Size
- 4 Contact Area
- 5 Preparation Technique
- 6 Preparation Time
- 7 Insertion Depth and Number of Insertions
- 8 Correlation with Displacement
- 9 Motor
- 10 Kinematic
- 11 Operative Motion
- 12 Rotational Speed and Pecking Speed
- 13 Lubricant
- 14 Experience
- 15 Metallurgy
- Line 147, page 12: “…canals can be lost during measuring…” Please fix this double space.
- Line 160, page 13: “…instrument (20).Instruments with…” Please fix this missing space.
- Line 178-180, page 13: “Thus, design had an impact on torque and force—shaping instruments with shorter pitch lengths produced greater torque and force, helix angle and rake angle affected torque and force, and higher cutting efficiencies were associated with lower torque and forces” There are too much torque and force in this sentence, please fix it in a more concise way.
- Line 238-239, page 16: “Sequential usage of instruments decreases torque and force for subsequently used instruments” Please avoid this repetition.
- Line 244-251, pg 16: “EdgeFile X7 (Edge Endo; Albuquerque, NM) had shorter preparation times and was associated with lower torque than ProTaper NEXT X1 (20)”. To optimize clinical performance, a NiTi rotary instrument should require low operative torque values and exhibit high resistance to torque at failure, or exhibit a wider “torque range” between the two values.
This extremely interesting and innovative concept could be better developed if we consider this article:
“Gambarini G, Miccoli G, D'Angelo M, Seracchiani M, Obino FV, Reda R, et al. The relevance of operative torque and torsional resistance of nickel-titanium rotary instruments: A preliminary clinical investigation. Saudi Endod J 2020;10:260-4.”
The EdgeFile X7 has a wider “torque range” when compared to PTN X1. This new concept could be a relevant innovation to match in vivo and in vitro studies, and to analyse this characteristic with a new and complete test. - Line 257-258, pg 16: “Thus, short pecking depths should be applied in clinical settings with great caution, particularly in the deepest part of the anal” Please fix this sentence.
- Line 324-325, pg 18: “Studies showed that lubricant reduced torque and force and that the aqueous type was superior to paste- or gel-type (53, 54)”. Although the review considers the articles published by July 2020, I suggest inserting this article to enrich the lubricants section even only in the discussion, to put more importance on this fundamental topic:
“Mazzoni, A.; Pacifici, A.; Zanza, A.; Giudice, A.D.; Reda, R.; Testarelli, L.; Gambarini, G.; Pacifici, L. Assessment of Real-Time Operative Torque during Nickel–Titanium Instrumentation with Different Lubricants. Sci.2020, 10, 6201.” - Line 345, page 18: “made of stainless steel (Gentlefile, MedicNRG, City Israel)” Please fix this double space.
Conclusions are concise and clear:
In Conclusions:
- Line 359, page 19: “deep insertion depth” Please avoid this repetition.
English is clear and easy to understand.
No improper citations are evidenced. When articles are cited, the journal format requires no space after comma. Please correct this in the text whenever it is present.
However, some improvements are mandatory before acceptance, you will find them in the attached file.
Round 2
Reviewer 2 Report
This research is under the scope of this journal; the topic is interesting for readers and this research deals with potentially significant knowledge to the field and an open new way for future studies.
The authors improved the quality of the manuscript after the reviewer's indications. Congratulations!!
Reviewer 3 Report
I believe it is now suitable for publication.